# Single-Cell RNA Sequencing Reveals Immuno-Oncology Characteristics of Tumor-Infiltrating T Lymphocytes in Photodynamic Therapy-Treated Colorectal Cancer Mouse Model

**DOI:** 10.3390/ijms241813913

**Published:** 2023-09-10

**Authors:** Eun-Ji Lee, Jang-Gi Choi, Jung Ho Han, Yong-Wan Kim, Junmo Lim, Hwan-Suck Chung

**Affiliations:** 1Korean Medicine Application Center, Korea Institute of Oriental Medicine (KIOM), Daegu 41062, Republic of Korea; jistr@kiom.re.kr (E.-J.L.); jang-gichoi@kiom.re.kr (J.-G.C.); hanjh1013@kiom.re.kr (J.H.H.); 2Dongsung Cancer Center, Dongsung Biopharmaceutical, Daegu 41061, Republic of Korea; thomas06@hanmail.net (Y.-W.K.); ijm15@ds-pharm.co.kr (J.L.); 3Korean Convergence Medical Science Major, University of Science and Technology (UST), KIOM Campus, Daegu 41062, Republic of Korea

**Keywords:** photodynamic therapy, single-cell RNA sequencing, cancer immunology, humanized PD-1 mice, tumor-infiltrating T cell

## Abstract

Photodynamic therapy (PDT) has shown promise in reducing metastatic colorectal cancer (CRC); however, the underlying mechanisms remain unclear. Modulating tumor-infiltrating immune cells by PDT may be achieved, which requires the characterization of immune cell populations in the tumor microenvironment by single-cell RNA sequencing (scRNA-seq). Here, we determined the effect of Chlorin e6 (Ce6)-mediated PDT on tumor-infiltrating T cells using scRNA-seq analysis. We used a humanized programmed death-1/programmed death ligand 1 (PD-1/PD-L1) MC38 cell allograft mouse model, considering its potential as an immunogenic cancer model and in combination with PD-1/PD-L1 immune checkpoint blockade. PDT treatment significantly reduced tumor growth in mice containing hPD-1/PD-L1 MC38 tumors. scRNA-seq analysis revealed that the PDT group had increased levels of CD8^+^ activated T cells and CD8^+^ cytotoxic T cells, but decreased levels of exhausted CD8^+^ T cells. PDT treatment also enhanced the infiltration of CD8^+^ T cells into tumors and increased the production of key effector molecules, including granzyme B and perforin 1. These findings provide insight into immune-therapeutic modulation for CRC patients and highlight the potential of PDT in overcoming immune evasion and enhancing antitumor immunity.

## 1. Introduction

Colorectal cancer (CRC) is a significant global health burden which requires the development of novel therapeutic approaches to improve patient outcomes. Immunotherapy has emerged as a promising strategy for cancer treatment as it harnesses the power of the immune system to recognize and eliminate tumor cells. Among the various immunotherapeutic modalities, immune checkpoint inhibitors targeting programmed cell death protein 1 (PD-1) and its ligand (PD-L1) have shown remarkable success in several cancer types, including CRC [1,2]. However, a considerable proportion of patients do not respond to these therapies, which indicates the need to identify alternative strategies to enhance antitumor immunity.

Photodynamic therapy (PDT) is a minimally invasive therapeutic approach that utilizes photosensitizing agents and light activation, resulting in the generation of reactive oxygen species and subsequent tumor destruction [3]. Chlorin e6 (Ce6), a second-generation photosensitizer, exhibits antitumor activities during photodynamic therapy [4]. Some studies have indicated that efficiently internalized Ce6 is enriched in the mitochondria and exhibits almost no cytotoxicity under dark conditions but strong photocytotoxicity resulting from reactive oxygen species (ROS) production during photodynamic therapy [5]. Recent studies have demonstrated an antitumor immune response for combinations containing PDT, which indicates an important role of the immune response for improved therapeutic results [6]. PDT induction of acute inflammation improves the tumor response rate and results in increased levels of inflammatory cytokines and neutrophil infiltration into photodamaged tumor cells [7]. Besides its direct cytotoxic effects, PDT induces an immune response that contributes to the regression of both primary and metastatic tumors [8]. The precise mechanism by which PDT exerts its immunomodulatory effects in CRC, particularly in the context of metastatic disease, remains poorly understood. Several studies have implicated a role for tumor-infiltrating immune cells in mediating the therapeutic response to PDT [9,10]. In addition, PDT modulates the tumor microenvironment (TME) and stimulates the immune response, making it a potential candidate for combination with immunotherapy [11,12]. In fact, recent studies have revealed the immunomodulatory properties of PDT therapy with respect to CD8^+^ T cell immunity in local and distant tumors [13,14]. Activated CD8^+^ T cells are cytotoxic T lymphocytes that are involved in eliminating tumor cells in the TME by releasing granule-associated enzymes, granzyme B (Gzmb), and granule exocytosis perforin1 (Prf1) [15,16].

Previously, PDT treatment developed by DongSung BioPharmaceuticals significantly reduced tumor growth in B16F10 melanoma allograft tumors and PANC02 pancreatic tumors in mice [17,18]. In this study, we examined the immuno-oncological characteristics of tumor-infiltrating T lymphocytes in a photodynamic therapy-treated CRC mouse model. We used a humanized PD-1/PD-L1 knock-in MC38 tumor model, considering its potential as an immunogenic cancer model and in combination with programmed death-1/programmed death ligand 1 (PD-1/PD-L1) immune checkpoint blockade. Ce6, a photosensitizer, was administered intravenously followed by laser irradiation at the tumor site. We hypothesized that Ce6-mediated PDT suppresses tumor growth while promoting an antitumor immune response.

Although, the advent of single-cell RNA sequencing (scRNA-seq) has revolutionized our understanding of cell heterogeneity and function within complex tissues, including tumors [19], few studies have examined the underlying mechanism of PDT using scRNA-seq. To obtain a comprehensive understanding of the immune landscape associated with the TME, we performed a scRNA-seq analysis of CD3^+^ T cells isolated from tumor tissues of PDT-treated mice. Through transcriptomic profiling and clustering analysis, we identified distinct T cell subsets and characterized their gene expression profiles. Furthermore, we compared the T cell populations between the control and PDT-treated groups to determine the effect of PDT on T cell composition and activation status.

In addition to scRNA-seq analysis, we examined the phenotypic changes in CD8^+^ T cells from humanized PD-L1 MC38 tumors after Ce6-mediated PDT treatment. Immunohistochemical staining was performed to assess T cell infiltration and activation markers, which provided insights into the enhanced antitumor immune response elicited by PDT. We also measured the secreted levels of key immune effector molecules in the serum to further validate the therapeutic efficacy of PDT.

Our study provides a comprehensive understanding of the immuno-oncological characteristics of tumor-infiltrating T lymphocytes in the context of Ce6-mediated PDT in CRC. The results have the potential to contribute to the development of novel combination therapies incorporating PDT and immunotherapy, ultimately improving the treatment outcomes for CRC patients.

## 2. Results

### 2.1. Ce6-Mediated PDT Induces Antitumor Effects in Mice Bearing Humanized PD-1/PD-L1 Knock-In MC38 Tumors

To determine whether Ce6-mediated PDT inhibits tumor growth in vivo, we established a humanized PD-1/PD-L1 CRC mouse model using MC38 cells (Figure 1A). When tumors reached approximately 100 mm^3^ in volume 11 days after s.c. administration, the mice were i.v. injected with Ce6 and then irradiated with a 660 nm laser at 200 mW/cm^2^ for 500 s at the site of the tumor. As shown in Figure 1B, treatment with Ce6-mediated PDT markedly suppressed tumor growth by 78% after 19 days compared with the control group. Tumor progression was significantly delayed in PDT-treated mice compared with the control group after tumor tissue excision on day 20 (Figure 1A,C). In addition, PDT therapy led to complete and long-lasting cancer cure for 19 days (Figure 1D). The body weight of the mice treated with PDT steadily increased compared with that of the control group, as observed throughout the experiments (Figure 1E). PDT treatment did not cause a significant change in spleen weight on day 20 (Figure 1F). GOT and GPT activity in the liver and UREA and CREA levels for kidney function were in the normal range (Figure 1G). The results indicate that Ce6-mediated PDT exhibits excellent antitumor effects with satisfactory safety and biocompatibility.

### 2.2. Single-Cell RNA-seq Analysis of T Cells in Tumor Tissues by Ce6-Mediated PDT Treatment

To establish comprehensive single-cell transcriptomes of the tumor-infiltrating T lymphocytes, we carried out single-cell RNA sequencing (scRNA-seq) of hPD-1-expressing CD3^+^ T cells isolated from hPD-L1-expressing MC38 tumor tissues of Ce6-mediated PDT-treated mice. Single-cell transcriptomes were established from 964 cells (123 for the control group and 841 for the PDT group) of sorted CD3^+^ T cells, which expressed *Cd3d*, *Cd3e,* and *Cd3g* in all subpopulations of the tumors. This resulted in 12,924 genes after quality control and filtering. The T cell subsets were divided into eight clusters by the highly expressed genes of the canonical T cell markers and the T cell types and visualized in two dimensions using UMAP (Figure 2A). The top 12 differentially expressed genes from the single-cell transcriptome data were revealed for each cluster of eight T cell subsets, including six CD8^+^ T cells and two CD4^+^ T cells (Figure 2B). Six clusters highly expressing the *Cd8a* and *Cd8b1* genes were assigned as CD8^+^ T cells: exhausted T cells expressing the gene *Klrc1* [20,21]; effector memory T cells expressing *Isg15*, *Itgb1*, and *Gzma* [22,23,24]; activated T cells expressing the activation marker and transcription factor *Irf8* [25,26] and the proliferation markers *Mki67*, *Stmn1*, *Top2a*, and *Cdk1* [27,28]; naïve T cells expressing *Ccr7* [29] and the transcription factors *Tcf7*, *Sell*, and *Lef1* [30]; memory T cells expressing *Ly6c2*, *Cd7*, *Plac8*, and *Itga1* [31,32,33,34,35]; and cytotoxic T cells expressing the cytotoxic markers *Prf1*, *Gzmb*, and *Nkg7* [36,37,38] and the transcription factor *Eomes* [39]. The CD4^+^ T cells comprised two clusters: a regulatory T cell cluster defined by the canonical genes *Foxp3*, *Il2ra*, *Maf*, *Ccr2*, and *Klrg1* [40,41,42,43] as well as an effector memory T cell cluster defined by the canonical genes *Icos*, *Rora*, *Cd81*, and *Ly6a* [44,45,46,47,48,49,50]. The integrated single-cell transcriptome analysis of the control and Ce6-mediated PDT treatment group of T cell subsets was visualized in two-dimensional space using UMAP (Figure 2C). In the control group, CD4^+^ regulatory T cells and CD4^+^ effector memory T cell distributions were higher among the eight T cell subpopulations. In the Ce6-mediated PDT group, more diversity was observed, including CD8^+^ effector memory T cells, CD8^+^ activated T cells, CD8^+^ naïve T cells, and CD8^+^ memory T cells. Notably, most of the CD8^+^ cytotoxic T cell subsets were distributed in the Ce6-mediated PDT group. The differential expression levels of T cell identity markers were represented in each T cell subpopulation using UMAP and dot plots, which included effector, activation, and central memory T cell marker genes (*Ctla4*, *Cd28*, *Cd69*, *Il7r*, *Itgal*, *Lef1*, *Tcf7*, and *Ccr7*) [51,52,53] and regulatory T cell marker genes (*Foxp3* and *Il2ra*) (Figure 2D,F). The expression of genes important for T cell regulation, differentiation, and function was presented for each T cell subpopulation using UMAP and a dot plot, including inhibitory molecules (*Lag3*, *Tigit*, and *Cx3cr1*) [54,55,56]; molecules in the resting states associated with regulatory T cells (*Ccr2* and *Tnfrsf18*) [57]; molecules associated with effector, activation, and central memory T cells (*Isg15*, *Icos*, *Rora*, *Lat*, *Cd81*, *Lck*, and *Plac8*) [58,59]; and cytotoxic molecules (*Nkg7*, *Gzmb*, *Gzmk*, and *Prf1*) (Figure 2E,G). Ce6-mediated PDT-treated mice exhibited a higher proportion of CD8^+^ cytotoxic T cells, CD8^+^ memory T cells, and CD8^+^ naïve T cells, but lower proportions of CD8^+^ exhausted T cells and CD4^+^ effector memory T cells compared with the control mice (Figure 2H). In the Ce6-mediated PDT group, the *Cd69* activation signature gene; the *Lck*, *Plac8*, and *Il7r* memory-like T cell markers; and the *Gzmb* and *Prf1* cytotoxicity signature genes were expressed at higher levels compared with the control group in all T cell subsets (Figure 2I). In contrast, the *Lag3*, *Havcr2*, and *Vsir* genes, which are associated with immunoglobulin suppression of T cell activation [60,61] and the distinct regulatory T cell defining genes, *Foxp3*, *Il2ra*, and *Tnfrsf18,* exhibited decreased expression in the Ce6-mediated PDT group compared with the control group for all T cell subsets. Overall, these findings indicate that more CD8^+^ activated T cells and CD8^+^ cytotoxic T cells induced by Ce6-mediated PDT treatment are the predominant populations that correlate with the antitumor immune response.

### 2.3. Characterizations of CD8^+^ T Cells in Humanized PD-L1 MC38 Tumors of PDT-Treated Mice

We previously categorized CD8^+^ T cells into six clusters, including activated, cytotoxic, effector memory, exhausted, memory, and naïve T cells highly expressing *Cd8a* and *Cd8b1* in eight T cell subsets. To further characterize the tumor-associated CD8^+^ T cells in the TME after Ce6-mediated PDT treatment, we identified genes associated with biomarkers and functions involved in tumor growth for each of the six CD8^+^ T cell subsets. Several genes, including *Cd28*, *Tnfrsf4*, *Ccna2*, *Cd226*, *Cd2*, *Psme2*, *Smc4*, *Lat*, and *Ube2c,* were strikingly different in CD8^+^ activated T cells among the subsets [62,63,64] (Figure 3A). Interestingly, we found that one CD8^+^ activated T cell subset expressed high levels of *Mki67*, which has proliferative potential, and other cell cycle markers, such as *Pclaf*, *Pcna*, *Cdk1*, *Stmn1*, *Top2a*, *Tuba1b*, and *Tyms* [65,66,67] (Figure 3B). One hallmark of tumor-infiltrating T cells is their ability to kill cancer cells. They express cytotoxic markers, including *PD-1*, *Gzmb*, *Gzmk*, and *Prf1* [68,69]. Our data indicated that *Gzmb*, *Gzmk*, *Prf1, Eomes*, *Irf8*, *Klrd1*, *Fyn*, *Nfatc1*, *Nkg7*, *Tnfrsf9*, and *Zap70* were highly expressed in the PD-1^+^ CD8^+^ cytotoxic T cell subset [70,71,72] (Figure 3C). *Epsti1*, *Isg15*, *Itgal*, *Lck*, *Ly6a*, *Ly6c2*, and *Ly6e,* which have putative roles in the activation of the CD8^+^ T cell immune response, were highly enriched in CD8^+^ effector memory T cells [73] (Figure 3D). Previously, we found that *Klrc1*, which has an inhibitory role by downregulating T cell immune function, was markedly expressed in CD8^+^ exhausted T cells. Moreover, *Cd7*, *Cxcr4*, *Itga1*, *Plac8*, and *Trgc1* were highly enriched in CD8^+^ memory T cells, whereas *Emb*, *Hspa1a*, *Lef1*, *Sell*, and *Tcf7,* transcriptional regulators of T cell development, were expressed in CD8^+^ naïve T cells (Figure 3E).

All six CD8^+^ T cell subsets showed considerable differences between the control and Ce6-mediated PDT groups (Figure 3F). The most striking observation was the accumulation of CD8^+^ cytotoxic T cells and CD8^+^ naïve T cells in the hPD-L1 MC38 tumors of the Ce6-mediated PDT-treated mice in contrast to the highly enriched CD8^+^ exhausted T cells in the tumors of the control mice. Consistently, our data showed that the *Havcr2*, *Lag3*, and *Vsir* genes, which are CD8^+^ exhausted T cell markers, were highly expressed in the control group (Figure 3H). In addition, *Cd69*, *Itgal*, *Lck*, *Cd2*, *Pcna*, *Emb*, *Ms4a4b*, *Plac8*, and *Ctla4*, which are specific for CD8^+^ T cell activation and development as well as CD8^+^ cytotoxic molecules, including *Gzmb* and *Nkg7*, were notably expressed in the PDT treatment group. We extracted three CD8^+^ T cell clusters, known to directly regulate antitumor immunity in the TME, including CD8^+^ activated T cells, CD8^+^ cytotoxic T cells, and CD8^+^ exhausted T cells. The infiltrated CD8^+^ activated T cells indicated high levels of the activation markers *Itgal*, *Cd69*, *Cd2*, *Lck*, *Pcna*, *Hmgb1*, and *Ctla4* in the PDT-treated mice (Figure 3G). Moreover, the infiltrated CD8^+^ cytotoxic T cells indicated high levels of *Gzmb* and *Nkg7* in the PDT-treated mice. In contrast, the infiltrated CD8^+^ exhausted T cells exhibited decreased expression of *Havcr2*, *Lag3*, and *Vsir* in the PDT treatment group. We further confirmed the genetic changes in tumor-associated CD4^+^ T cells of effector memory T cells and regulatory T cells in the TME after Ce6-mediated PDT treatment. A subset of CD4^+^ effector memory T cells highly enriched in *Ctla4*, *Il7r*, *Jund*, *Ly6e*, and *Cxcr4* were found in the PDT treatment group [74], and a CD4^+^ regulatory T cell cluster highly expressing *Il2ra*, *Tigit*, *Cd74*, *Ctss*, *Serpinb6a*, and *Serpinb6b* was identified in the control group [75,76,77] (Figure 3I). Collectively, these results show that Ce6-mediated PDT treatment resulted in the expression of genes that are important targets for immunotherapy and promote antitumor immunity through CD8^+^ T cell infiltration.

### 2.4. Ce6-Mediated PDT Enhances Antitumor Immunity in Humanized PD-L1 MC38 Tumors

To further confirm the contribution of T cells in mediating antitumor immunity, we performed immunohistochemical staining of an independent set of humanized PD-L1 MC38 allograft tumor tissues. As shown in Figure 4A,B, Ce6-mediated PDT significantly increased T cell infiltration by upregulating CD3, CD4, and CD8 expression compared with the control group. In particular, CD8^+^ T cell in tumors treated with Ce6-mediated PDT exhibited a stronger activation phenotype, including the production of Gzmb and Prf1, compared with that in the control group. This indicates the ability of Ce6-mediated PDT to promote CD8^+^ T cell infiltration into tumors. We also measured the secretion of Gzmb, IFN-γ, and IL-2 in mouse serum, and they were enhanced by Ce6-mediated PDT (Figure 4C–E). The above results indicate that Ce6-mediated PDT induces antitumor immunity by enhancing the abundance of CD8^+^ T cell infiltration in CRC tumors of mice bearing humanized PD-1/PD-L1 knock-in MC38 tumors.

## 3. Discussion

In this study, we determined the antitumor effect and immune responses induced by Ce6-mediated PDT in a humanized PD-1/PD-L1 colorectal cancer (CRC) mouse model bearing MC38 tumors. Our results demonstrate that Ce6-mediated PDT effectively inhibited tumor growth and delayed tumor progression compared with the control group. Interestingly, T cell-deficient athymic nude mice showed no response to PDT in an orthotopic mouse model of pancreatic cancer bearing large tumors (35–65 mm^3^) [78]. Since PDT treatment was administered when the tumor size reached 100 mm^3^, the tumor suppression effect of PDT may not involve direct tumor killing via ROS generation, but indirect killing via immune cells, such as T cells. Although we cannot compare the antitumor effect of PDT with that of other studies because the tumor models and PDT treatment were different, 78% inhibition of tumor growth by PDT represents a dramatic effect. MC38 cells are immunogenic and exhibit high expression of the co-stimulatory molecules CD40 and CD86 on the surface of dendritic cells compared with CT26 colon cancer cells [79]; thus, PDT is more effective against MC38 tumor-bearing models compared with CT26 [80] or LLC [81] tumor-bearing models. We hypothesize that immunogenic tumors will be more sensitive to PDT treatment in the clinic based on these preclinical results.

To gain insight into the immune response associated with Ce6-mediated PDT, we performed a single-cell RNA sequencing (scRNA-seq) analysis of tumor-infiltrating T lymphocytes in the MC38 tumor tissue of treated mice. By characterizing the transcriptomes of CD3^+^ T cells, we identified eight distinct T cell subsets, including six CD8^+^ T cell subsets and two CD4^+^ T cell subsets. Notably, Ce6-mediated PDT resulted in higher proportions of CD8^+^ activated T cells, CD8^+^ cytotoxic T cells, and CD8^+^ naïve T cells, but reduced numbers of CD8^+^ exhausted T cells and CD4^+^ effector memory T cells compared with the control group. Further characterization of CD8^+^ T cell subsets revealed specific gene expression patterns associated with tumor growth and the immune response. CD8^+^ activated T cells exhibited higher expression of genes, including *Mki67*, *Pclaf*, *Pcna*, *Cdk1*, *Stmn1*, *Top2a*, *Tuba1b*, and *Tyms*, related to cell cycle control and proliferation, indicating their active and proliferative state [27,28,65,66,67], as well as the activation markers *Irf8*, *Cd28*, *Tnfrsf4*, *Ccna2*, *Cd226*, *Cd2*, *Psme2*, *Smc4*, *Lat*, and *Ube2c* [25,26,62,63,64]. CD8^+^ cytotoxic T cells had elevated expression of the cytotoxic markers *PD-1*, *Gzmb*, *Gzmk*, *Prf1*, *Nkg7*, *Eomes*, *Irf8*, *Klrd1*, *Fyn*, *Nfatc1*, *Tnfrsf9*, and *Zap70,* supporting their killing function against cancer cells [38,39,68,69,70,71,72,82]. CD8^+^ effector memory T cells exhibited enrichment of genes, including *Epsti1*, *Isg15*, *Itgal*, *Lck*, *Ly6a*, *Ly6c2*, *Ly6e*, *Itgb1*, and *Gzma,* which are associated with T cell activation and immune response [22,23,24,73], whereas CD8^+^ exhausted T cells showed upregulation of inhibitory molecules, including *Klrc1* [20,21]. CD8^+^ memory T cells (*Ly6c2*, *Cd7*, *Plac8*, *Itga1, Cxcr4*, and *Trgc1*) [31,32,33,34,35] and CD8^+^ naïve T cells (*Ccr7*, *Tcf7*, *Sell*, *Lef1*, *Emb*, and *Hspa1a*) [29,30] exhibited unique gene expression profiles related to memory development and T cell differentiation, respectively.

A comparison between the control and PDT-treated groups revealed distinct alterations in the composition and activation status of CD8^+^ T cell subsets. The proportion of CD8^+^ naïve T cells in the PDT group was two-fold higher than that of the control group (7% in Control vs. 14% in PDT). Naïve CD8^+^ T cells circulate in the bloodstream and lymphoid organs, constantly scanning for the presence of their specific antigen. When a naïve CD8^+^ T cell encounters antigen-presenting cells (APCs) presenting its specific antigen, the T cell receptor (TCR) on the CD8^+^ T cell recognizes the antigen–MHC complex. This interaction, along with co-stimulatory signals from the APC, triggers the activation of the naïve CD8^+^ T cell. Upon activation, naïve CD8^+^ T cells undergo clonal expansion, resulting in the generation of a large population of effector CD8^+^ T cells [83]. They further differentiate into memory and cytotoxic cells [84]. Ce6-mediated PDT resulted in a high accumulation of CD8^+^ cytotoxic T cells, which are a subset of activated CD8^+^ T cells that have cytotoxic capabilities (0% in Control vs. 9% in PDT) [85]. In the PDT-treated group, among the CD8^+^ T cell types directly regulating antitumor immunity in the TME, genes associated with CD8^+^ activated T cells (*Itgal*, *Cd69*, *Cd2*, *Lck*, *Pcna*, *Hmgb1*, and *Ctla4*) and CD8^+^ cytotoxic T cells (*Gzmb* and *Nkg7*) were highly expressed, whereas CD8^+^ exhausted T cell genes (*Havcr2*, *Lag3*, and *Vsir*) exhibited reduced expression. Cytotoxic T cells recognize antigens displayed on the surface of target cells through their TCRs. Once the cytotoxic T cell recognizes the antigen, it releases cytotoxic molecules, such as perforin and granzymes, which induce apoptosis of the target cells [86].

Immunohistochemical staining of tumor tissues further supported the enhanced infiltration of T cells following Ce6-mediated PDT. Increased expression of CD3, CD4, and CD8 was observed, with CD8^+^ T cells showing an activated phenotype characterized by the production of Gzmb and Prf1. These results indicate that Ce6-mediated PDT not only promotes CD8^+^ T cell infiltration, but also enhances their functional activation in the TME. Furthermore, the secretion of Gzmb, IFN-γ, and IL-2 in the serum of treated mice was elevated, indicating the systemic activation of antitumor immune responses by Ce6-mediated PDT.

We performed scRNA-seq using only CD3^+^ T cells to examine the detailed changes in T cells by PDT. While there are many types of immune cells in the TME, such as natural killer cells, dendritic cells, macrophages, and myeloid-derived suppressor cells, the changes in these cells’ gene expression and cell populations caused by PDT were not examined in this study. To explore the changes in all immune cells in the TME, scRNA-seq should be performed on CD45^+^ cells.

In summary, our study demonstrates that Ce6-mediated PDT effectively induces antitumor immunity in a humanized PD-1/PD-L1 MC38 tumor model. The treatment resulted in the inhibition of tumor growth, increased the infiltration of activated and cytotoxic CD8^+^ T cells, and enhanced systemic immune response. These findings highlight the potential of Ce6-mediated PDT as a promising therapeutic strategy for CRC and provide insight into the underlying mechanisms of its antitumor effects.

## 4. Materials and Methods

### 4.1. Preparation of Ce6

Ce6 complexed with polyvinylpyrrolidone (PVP) (Ce6-PVP complex), designated Phonozen^®^, was obtained from DongSung BioPharmaceuticals (Daegu, Republic of Korea) and synthesized as described previously [17,18]. Briefly, *Spirulina platensis* (SP) was provided by the East India Distilleries Parry (EID Parry, Chennai, India). The dried SP (10 kg) was used to extract chlorophyll a by stirring in 100 L of EtOH for 8 h in a N_2_ atmosphere. The mixture was filtered and washed with 30 L of hexane and 8.5 L of distilled water. The resulting chlorophyll a-containing filtrate was treated with 1N HCl to obtain pheophytin a, which was purified using liquid–liquid extraction and dissolved in acetone. Anhydrous sodium sulfate was added to the pheophytin a solution, which was then refluxed at 65–72 °C after filtration, followed by a pH adjustment using 1N NaOH. The resulting Ce6 precipitate was filtered, washed with acetone, and vacuum-dried at 40 °C. The solid Ce6 and PVP were mixed at a weight ratio of 1:1, filtered through a 0.22 μm membrane filter, and the sterile-filtered Ce6-PVP complex was lyophilized to prepare a powder (yield: 95%).

### 4.2. Humanized PD-1/PD-L1 Knock-In Colorectal Cancer Mouse Model

The 6–8-week-old male humanized PD-1 knock-in mice were used to establish a humanized PD-L1 colorectal cancer (CRC) allograft model, a genetically modified C57BL/6J mouse expressing the human PD-1 protein (hPD-1 mice). The hPD-1 mouse has a chimeric PD-1 with a human extracellular domain, a mouse transmembrane domain, and a mouse intracellular domain. So, the mouse preserves the target–ligand interaction and fully functional mouse immune system. Murine CRC MC38 cells stably expressing human PD-L1 (hPD-L1 MC38 cells) were purchased from the Shanghai Model Organisms Center (Shanghai, China). All procedures were performed with the approval of the Institutional Animal Care and Use Committee of the Korea Institute of Oriental Medicine (KIOM) (approval number: KIOM-D-22-103). All mouse experiments were performed in accordance with the Guide for the Care and Use of Laboratory Animals of the National Institutes of Health of Korea.

To determine the therapeutic effect of PDT, hPD-L1 MC38 cells (3 × 10^5^/200 μL) were subcutaneously injected into the right dorsal skin of each male hPD-1 mouse. Tumor diameters were measured using digital calipers (Hi-Tech Diamond, Westmont, IL, USA), and tumor volume was calculated according to the formula: (length × width^2^)/2. When the tumor volumes reached approximately 100 mm^3^ (day 11 post inoculation), the hPD-1/PD-L1 MC38 tumor allograft mice were divided randomly into control and Ce6-mediated PDT groups (8 per group). Mice were i.v. injected with saline and 2.5 mg/kg Ce6 solution in the control and Ce6-mediated PDT treatment groups, respectively. Three hours after Ce6 solution administration, the mice were irradiated with a 660 nm laser diode (LEMT, Minsk, Belarus) at a dose of 100 J/cm^2^ (irradiation fluence rate: 200 mw/cm^2^; irradiation time: 500 s) at the tumor site under anesthesia. Tumor size and body weight were measured twice a week. On day 20, the mice were sacrificed at the time indicated for the antitumor immunological experiments and scRNA-seq analysis.

### 4.3. Blood Biochemistry

Serum was collected into blood collection tubes (BD Biosciences, San Jose, CA, USA) by cardiac puncture of hPD-1/PD-L1 MC38 tumor allograft mice treated with Ce6-mediated PDT. The levels of glutamic oxalacetic transaminase (GOT, IU/L), glutamic pyruvate transaminase (GPT, IU/L), UREA (mg/dL), and creatinine (CREA, mg/dL) were measured using an XL 200 biochemical analyzer (Erba Lachema s.r.o, Mannheim, Germany). 

### 4.4. Isolation of Tumor-Infiltrating T Cells

Tumor tissues from hPD-1/PD-L1 MC38 tumor allograft mice treated with Ce6-mediated PDT were isolated and cut into small pieces. Using a gentleMACS Dissociator (Miltenyi Biotec, Auburn, CA, USA), the minced tumor tissues were dissociated into single-cell suspensions by combining mechanical tumor dissociation with enzymatic treatment in a gentleMACS C Tube (Miltenyi Biotec). Dissociated tumor tissues were incubated in complete medium with 1 mg/mL collagenase, 1 mg/mL trypsin inhibitor, and 0.1 mg/mL DNase for 40 min at 37°C using a MACSmix Tube Rotator (Miltenyi Biotec) for enzymatic digestion with a Tumor Dissociation Kit (Miltenyi Biotec). The tumor cell suspension was placed into a 50 mL tube in a cell strainer (SPL Life Sciences, Pochen, Republic of Korea), and the tumor cell suspension was filtered and pulverized to obtain a single-cell suspension. hPD-1/PD-L1 MC38 tumor-infiltrating CD3^+^ T cells were isolated in polystyrene round-bottom tubes by immunomagnetic negative selection using the Mouse CD3^+^ T Cell Isolation Kit (STEMCELL Technologies, Vancouver, BC, Canada). Before single-cell RNA-seq, dead cells were removed using the Dead Cell Removal (Annexin V) kit (STEMCELL Technologies). Isolated mouse CD3^+^ T cells with a viability higher than 90% were used for BD Rhapsody single-cell RNA-seq.

### 4.5. Single-Cell RNA Sequencing

According to the protocol described in the BD Rhapsody Express Single-Cell Analysis System User Guide (BD Biosciences, San Jose, CA, USA), the single-cell suspensions (approximately 1 × 10^5^ cells; described in Cell Sorting for Single-Cell Labeling) were washed with BD Pharmingen Stain Buffer (BD Biosciences). The prepared single-cell suspension was labeled in a Sample Tag tube, which had a sample oligo barcode conjugated to an Anti-Mouse CD45, Clone 30-F11 antibody, of the BD Mouse Immune Single-Cell Multiplexing Kit (BD Biosciences) at RT for 20 min. The labeled cell suspension was transferred to a 5 mL polystyrene Falcon tube (Corning, New York, NY, USA) and washed with BD Pharmingen Stain Buffer (BD Biosciences). The labeled cell pellets were resuspended with cold Sample Buffer from the BD Rhapsody Cartridge Reagent Kit (BD Biosciences) for single-cell capture, and 3 × 10^4^ cells were captured with the BD Rhapsody Single-Cell Analysis System, following the manufacturer’s protocol (BD Biosciences). The labeled single cells and Cell Capture Beads were loaded into a cartridge and captured with the BD Rhapsody Express Single-Cell Analysis System (BD Biosciences). The cell capture beads were washed with cold Bead Wash Buffer (BD Biosciences) and processed for cDNA synthesis using the BD Rhapsody cDNA Kit (BD Biosciences), according to the manufacturer’s instructions.

Single-cell libraries were prepared using the BD Rhapsody whole transcriptome analysis (WTA) amplification kit (BD Biosciences) according to the mRNA Whole Transcriptome Analysis and Sample Tag Library Preparation protocol (BD Bioscience). For the WTA library, cDNA was prepared by random priming and extension (RPE) and RPE amplification, followed by index PCR. For the sample tag library, cDNA was processed by nested PCR (PCR 1 and PCR 2), followed by index PCR. Purified WTA and sample tag libraries were quantified by qPCR according to the qPCR Quantification Protocol Guide (KAPA) and validated using an Agilent Technologies 4200 TapeStation (Agilent Technologies, Carpinteria, CA, USA). Libraries were then pooled and sequenced using the HiSeq platform (Illumina, San Diego, CA, USA), resulting in 150 bp paired-end reads. The sequencing depth of the WTA library was approximately 20,000 reads/cell, and the sample tag library was approximately 120 reads/cell. 

### 4.6. Single-Cell RNA Sequencing Data Processing

FastQC was used to assess the quality and basic statistics of the raw sequencing data (FASTQ, https://www.bioinformatics.babraham.ac.uk/projects/fastqc/, accessed on 2 March 2023). Following quality control, sequencing data were processed using the BD Rhapsody WTA Analysis Pipeline v1.10.1 and the human reference genome (GRCh38). The gene expression matrices generated by the BD Rhapsody WTA Analysis Pipeline v1.10.1 represent the number of unique molecular identifiers for each gene and cell index.

For downstream analysis, we used R v4.0.3 [87] and Seurat v3.2.2 [88]. Low-quality cells were eliminated by removing the cells expressing more than 20% of the mitochondrial gene. The scRNA-seq data were normalized using the SCTransform program, which identifies and scales the expression values of highly variable genes. Seurat’s RunPCA function was used to run PCA on the preprocessed matrix and the top 16 principal components were used for subsequent clustering analysis. To identify different cell clusters, we clustered cells based on their gene expression profiles using an SNN modularity optimization approach with a resolution of 0.4 using the “FindClusters” tool. Using the UMAP technique, Seurat’s RunUMAP function was used to build a two-dimensional picture of the clustered cells. Expression levels of the marker genes were used to achieve cluster annotation.

### 4.7. Immunohistochemistry

The tumor tissues were harvested, fixed with 10% formalin (Sigma-Aldrich, Burlington, MA, USA), and paraffin-embedded. The explant tumor tissue sections were cut into slides at 10 μm thickness. The tumor tissue sections were deparaffinized and rehydrated. Heat-induced epitope retrieval was carried out in a pressure cooker using Retrieval solution (10 mM citrate buffer, pH 6.0; Agilent Technologies). Endogenous peroxidase activity was inhibited for 20 min at RT with 3% H_2_O_2_ in methanol. The tumor tissue sections were immunostained with a primary antibody against CD3 (#16669, Abcam, Cambridge, UK), CD4 (#183685, Abcam), CD8 (#98941, Cell Signaling Technology, CST, Danvers, MA, USA), Gzmb (#46890, CST), and Prf1 (#31647, CST) and incubated overnight at 4 °C after washing with Tris-buffered saline (TBS). The paraffin sections were visualized with a DAKO Envision kit (Agilent Technologies) after being washed with TBS containing 0.05% Tween-20 (TBS-T). The paraffin sections were counter-stained with Mayer’s hematoxylin for histological investigation of tumor tissues. Images were captured with an Olympus BX53 microscope and a Fujifilm XC10 microscopic digital camera (Tokyo, Japan). IHC images were measured using the Color Histogram function of the Image J software program.

### 4.8. Granzyme B Measurement

Granzyme B (Gzmb) in mouse serum was measured by a sandwich ELISA (#88-8022, Thermo Fisher Scientific, Waltham, MA, USA), according to the manufacturer’s instructions. Briefly, the anti-Gzmb antibody diluted in PBS was coated onto 96-well plates (Corning) and incubated overnight at 4 °C. The plates were rinsed with PBS-T and blocked for 1 h at RT. Each well was treated with a biotin-conjugated antibody and Avidin-coupled horseradish peroxidase and then incubated at RT for 1 h and 30 min, respectively. The relative absorbance was measured at 450 nm with a SpectraMax i3 microplate reader (Molecular Devices, San Jose, CA, USA).

### 4.9. IFN-γ or IL-2 Measurement

IFN-γ (#555138) or IL-2 (#555148) levels in mouse serum were measured using a sandwich ELISA (BD Biosciences), according to the manufacturer’s instructions. Briefly, anti-IFN-γ or anti-IL-2 antibody was diluted in 0.1 M sodium carbonate buffer (pH 9.5) and coated onto 96-well plates (Corning) and incubated overnight at 4 °C. At RT, the plates were rinsed with PBS-T and blocked with PBS containing 10% (*v/v*) fetal bovine serum (FBS, GE Healthcare Life Sciences, Chicago, IL, USA) for 1 h. Each well was treated for 1 h at RT with biotinylated antibody and a streptavidin–horseradish peroxidase conjugate. The relative absorbance was measured at 450 nm with a SpectraMax i3 microplate reader (Molecular Devices).

### 4.10. Statistical Analysis

Statistical analyses were performed using GraphPad Prism v9 (GraphPad Software, Inc., La Jolla, CA, USA) and R v4.0.3. Statistical analyses were applied to biologically independent mice or technical replicates for each experiment. The normal distribution of the in vivo data (Control vs. PDT) was checked before performing a Student’s *t*-test and mean ± standard deviation (SD). The difference in mean values was analyzed by a two-tailed Student’s *t*-test, which was used for comparisons between two independent groups, as indicated. Error bars were reported as the means ± SDs. The significance level was denoted as * *p* < 0.05, ** *p* < 0.01, *** *p* < 0.001, and **** *p* < 0.0001. Survival curve comparison was performed using the log-rank (Mantel–Cox) test.

## 5. Conclusions

In conclusion, Ce6-mediated PDT resulted in direct hPD-L1-expressing CRC cell death and stimulated the systemic hPD-1-expressing T cell immune response in CRC tumors. Through scRNA-seq analysis, we identified the proportion of T cell composition and differences in gene expression associated with the T cell components in CRC tumors treated with Ce6-mediated PDT. In comparison with the control group, Ce6-mediated PDT showed T cell infiltration into tumors and a higher proportion of CD8^+^ cytotoxic T cells, CD8^+^ memory T cells, and CD8^+^ naïve T cells in the TME. The expression of characteristic marker genes of CD8^+^ activated T cells and CD8^+^ cytotoxic T cells was higher in the Ce6-mediated PDT group, and gene expression associated with CD8^+^ exhausted T cells was lower compared with the control group. In addition, Ce6-mediated PDT effectively inhibited hPD-L1 MC38 tumor growth by enhancing tumor-infiltrating CD8^+^ T cells and Gzmb and Prf1 release in the TME and augmenting the antitumor immune response in mice bearing hPD-1/PD-L1 MC38 tumors. Importantly, our scRNA-seq dataset provides a better understanding of the tumor-infiltrating T lymphocyte mechanism associated with PDT-mediated antitumor therapy in the clinic. Therefore, our results may provide a valuable resource for human T cell immunity in PDT therapeutic immunotherapies directed at the hPD-1/PD-L1 blockade in patients with CRC based on a comprehensive examination of the T cell immune profile in the TME.

## Figures and Tables

**Figure 1 ijms-24-13913-f001:**
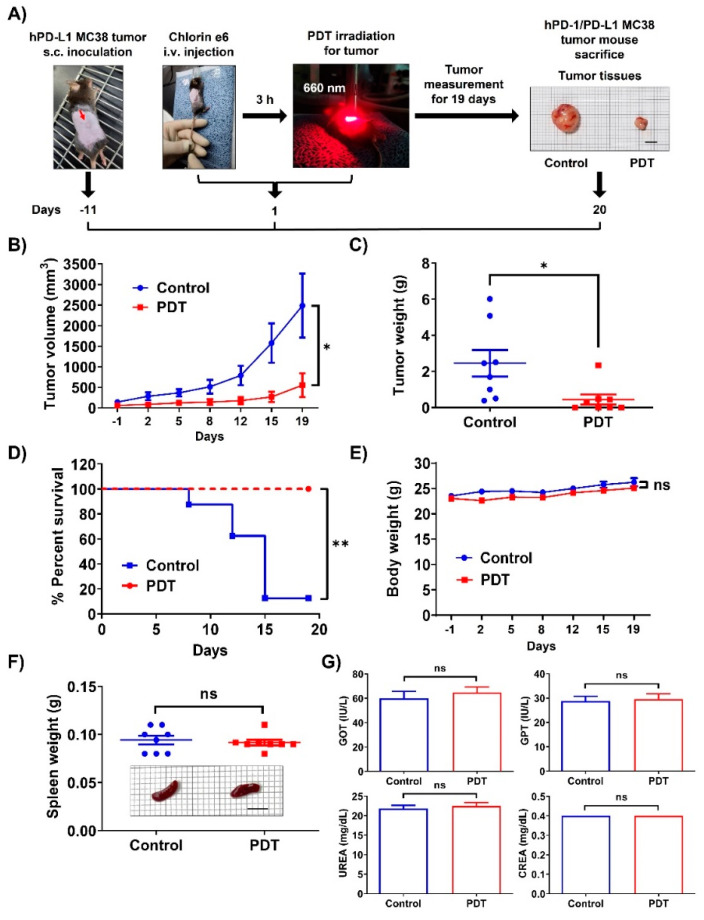
Tumor growth of hPD-L1 MC38 colorectal cancer in PDT-treated mice in vivo. (**A**) Scheme of the experimental steps to determine the antitumoral immune response triggered by Ce6-mediated PDT. Image of the incised tumor tissues in the control and PDT-treated mice on day 20 (bar indicates 1 cm). (**B**) Changes in tumor volume each day after PDT treatment. The statistical significance level was only shown for the last experimental day (day 19). (**C**) Tumor weight after tumor tissue excision on day 20. Dots represents individual values. (**D**) Survival curve of the control and PDT-treated mice. (**E**) Changes in body weight each day following PDT treatment. (**F**) Spleen weight after euthanization on day 20. Image of the incised spleen tissues in the control and PDT-treated mice on day 20 (bar indicates 1 cm). (**G**) Blood analysis of GOT, GPT, UREA, and CREA levels in sera performed in mice bearing hPD-1/PD-L1 MC38 tumors at the end of the experiment. Error bars represent the means ± SDs. * *p* < 0.05, ** *p* < 0.01.

**Figure 2 ijms-24-13913-f002:**
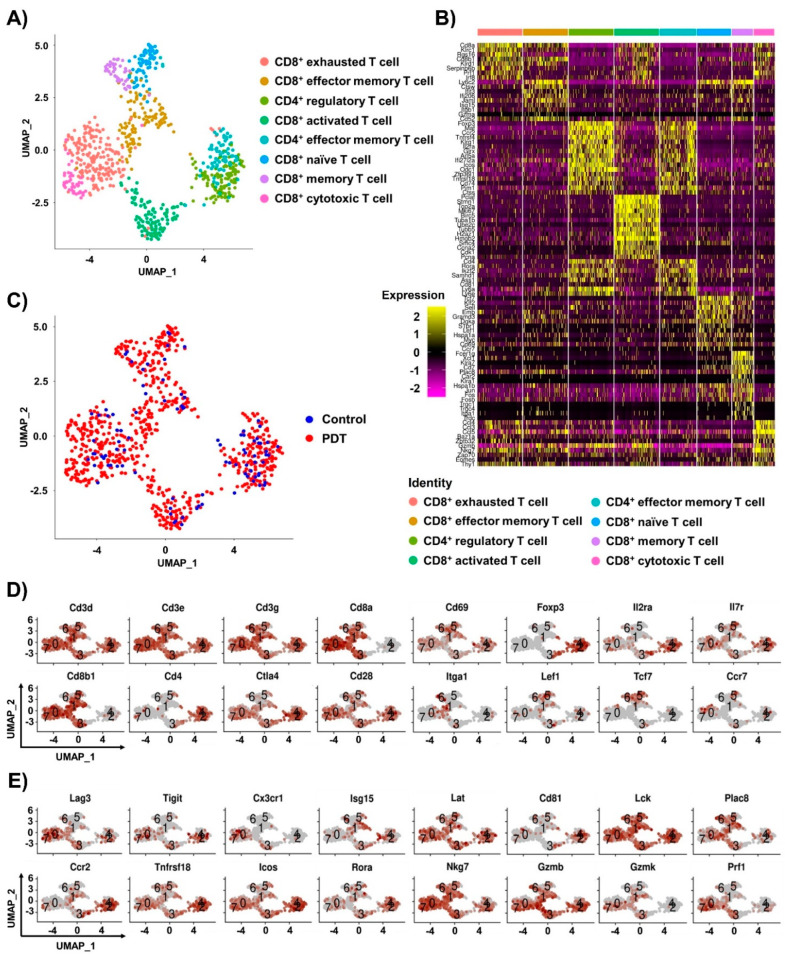
Changes in the T cell composition of tumors by PDT treatment through single-cell analysis. (**A**) UMAP plot displaying T cells in tumors. Cells are colored based on cell-type annotations. (**B**) Heatmap displaying the expression of the top differentially expressed genes in each T cell cluster of tumors. (**C**) UMAP plot displaying T cells from the control (blue) and PDT (red) groups, as indicated in the figure key. (**D**,**E**) UMAP plots showing the expression of the T cell type identity markers (**D**) and T cell functional markers (**E**). Numbers in the UMAPs indicate the T cell clusters (0, CD8^+^ exhausted T cells; 1, CD8^+^ effector memory T cells; 2, CD4^+^ regulatory T cells; 3, CD8^+^ activated T cells; 4, CD8^+^ effector memory T cells; 5, CD8^+^ naïve T cells; 6, CD8^+^ memory T cells; 7, CD8^+^ cytotoxic T cells). (**F**,**G**) Dot plots depicting the expression of T cell identity markers (**F**) and functional markers (**G**) with cell types. The color gradient of dots represents average expression; point sizes represent the percent expressed for the T cell markers. (**H**) Dot plot showing the T cell proportions of each T cell subtype in the control and PDT-treated groups. Cells are colored based on cell-type annotations, and proportions are indicated by the size of the dots. (**I**) Violin plots showing the expression of differentially expressed genes in T cells from tumors of the control (blue) and PDT (red) groups, as indicated in the figure key. Violin plot showing the log-normalized expression values extracted from the ‘data’ slot of the ‘SCT’ assay. Gene expression in (**A**–**G**), and (**I**) presents SCTransformed normalized values. UMAP; uniform manifold approximation and projection.

**Figure 3 ijms-24-13913-f003:**
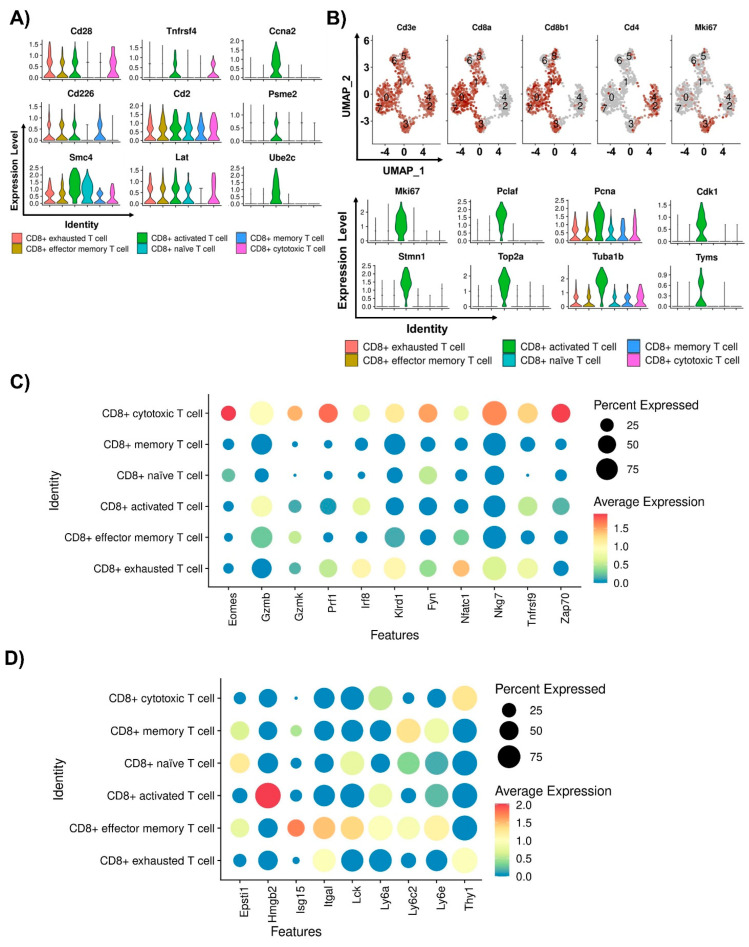
Characterization of tumor-infiltrating CD8^+^ T cells in PDT-treated mice through single-cell analysis. (**A**) Violin plots depicting the expression of representative marker genes of CD8^+^ activated T cells among different CD8^+^ T cell subtypes. (**B**) UMAP plots of T cells expressing select proliferation marker genes. Numbers in the UMAPs indicate the T cell clusters (0, CD8^+^ exhausted T cells; 1, CD8^+^ effector memory T cells; 2, CD4^+^ regulatory T cells; 3, CD8^+^ activated T cells; 4, CD8^+^ effector memory T cells; 5, CD8^+^ naïve T cells; 6, CD8^+^ memory T cells; 7, CD8^+^ cytotoxic T cells). Violin plots depicting the expression of proliferation marker genes of CD8^+^ activated T cells among different CD8^+^ T cell subtypes. (**C**,**D**) Dot plots depicting the expression of representative marker genes of CD8^+^ cytotoxic T cells (**C**) and CD8^+^ effector memory T cells (**D**) among different CD8^+^ T cell subtypes. (**E**) Violin plots depicting the expression of representative marker genes of CD8^+^ exhausted T cells, CD8^+^ memory T cells, and CD8^+^ naïve T cells among different CD8^+^ T cell subtypes. (**F**) Pie charts showing the proportion of CD8^+^ T cell subtypes in the control and PDT groups. (**G**) Dot plots depicting the expression of genes differentially expressed in CD8^+^ activated T cells, CD8^+^ cytotoxic T cells, and CD8^+^ exhausted T cells in tumors of the control (blue) and PDT (red) groups, as indicated in the figure key. (**H**) Violin plots showing the expression of differentially expressed genes in CD8^+^ T cells in tumors of the control (blue) and PDT (red) groups, as indicated in the figure key. (**I**) Violin plots showing the expression of differentially expressed genes in CD4^+^ regulatory T cells and CD4^+^ effector memory T cells in tumors of the control (blue) and PDT (red) groups, as indicated in the figure key. Violin plot showing the log-normalized expression values extracted from the ‘data’ slot of the ‘SCT’ assay. Gene expression in (**A**–**H**), and (**I**) presents SCTransformed normalized values. The color gradients of dots in C, D, and H represent the average expression; point sizes represent the percent expressed for the CD8^+^ T cell markers.

**Figure 4 ijms-24-13913-f004:**
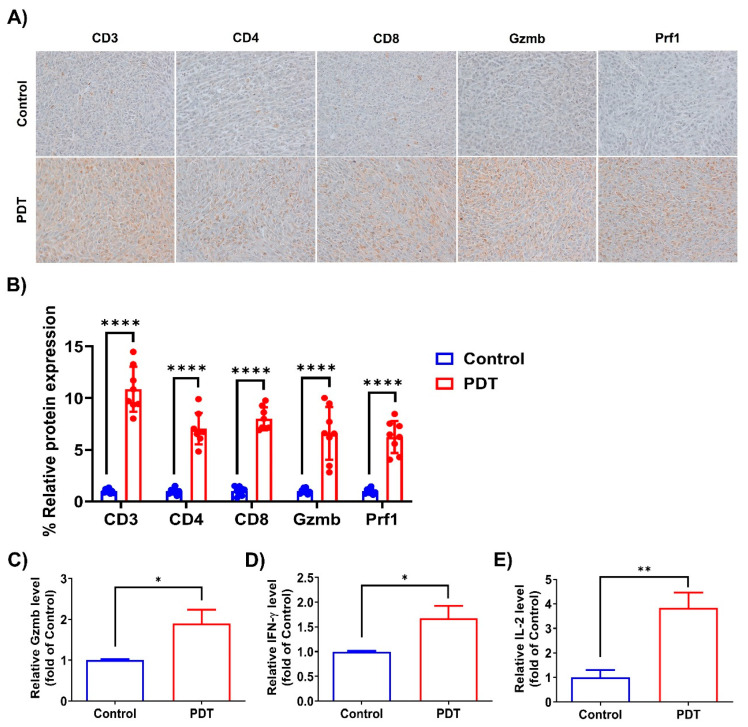
T cell immune effect following PDT treatment in mice bearing hPD-1/PD-L1 MC38 tumors. (**A**) IHC of CD3, CD4, CD8, Gzmb, and Prf1 was performed in tumor tissues at the end of the experiment (image size represents ×400). (**B**) Quantitation of CD3, CD4, CD8, Gzmb, and Prf1 expression in tumor tissues. Relative protein expression was quantitated using the ImageJ software. Dots represents individual values. (**C**–**E**) Gzmb (**C**), IFN-γ (**D**), and IL-2 (**E**) levels in mouse sera isolated on day 20 post-PDT treatment. The assays were performed in triplicate at each concentration. Error bars represent the means ± SDs. * *p* < 0.05, ** *p* < 0.01, **** *p* < 0.0001.

## Data Availability

The original contributions presented in the study are included in the article. Further inquiries can be directed to the corresponding author.

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
