# Peer review of "Single-Cell RNA Sequencing Reveals Immuno-Oncology Characteristics of Tumor-Infiltrating T Lymphocytes in Photodynamic Therapy-Treated Colorectal Cancer Mouse Model"

_ijms, 2023, doi:10.3390/ijms241813913_

Round 1

Reviewer 1 Report

My major concern is the complete lack of applicability of the study. I cannot see any real possibility to adapt it to clinical setting - how would one perform phototherapy of human CRC?

Major
1. Why only PD-1+ cells were used for sc RNAseq? Or was it as suggested in methods total CD3? If so, then why only cytotoxic T cells were analysed? How about T helpers, gamma delta T cells, iNKT or MAIT?
2. Typically mouse effector T cells are CD44+ - this is contrary to what authors suggest. Similarly, CD69 is one of typical activation markers - authors claim it is highly expressed on naive, but virually absent from activated T cells.
3. Figure 3H - No expression of granzyme B in control?
4. Figure 3. Colour code for average expression is unclear. What does it mean?
5. Clone, manufacturer and cat. number for each antibody should be clearly listed. Similarly, cat. numbers for ELISA kits.
6. Why student T test and mean/SD were used? First, data distribution should be tested and then appropriate further tests should be selected.
7. Authors fail to mention ANY limitations of the study while there are multiple...

Minor
1. Line 99: mW instead of mw
2. "immunomagnetic magnet"???
3. Pharmingen, not "Pharningen"

Professional proof-reading is highly advised

Reviewer 2 Report

The manuscript is very well written and organized. The topic is very relevant and worth of investigation; however, some questions need to be addressed before publication:

-       Have the authors investigated the effects of the laser without the photosensitizer?

-       How was the photosensitizer concentration determined?

-       How many PDT sessions were applied?

-       In the text is not clear when all the analysis were performed. Were they performed 20 days after 1 PDT application? Why did the authors decide to choose this timepoint?

-       Figure 1 B shows that PDT delayed tumor growth when compared to control rather than inhibited. Did the authors monitor tumor growth for a longer time, more than 20 days?

-       The authors mention that PDT might be more effective on immunogenic tumors. How would you describe the potential translation of this research in terms of the population affected by CRC?

Round 2

Reviewer 1 Report

I am mostly satisfied with the responses of the authors. My major concerns were sufficiently and clearly addressed.

Only the statistical comment requires 2nd look. As mentioned, at first authors have to analyse data distribution (normal vs other than normal) and then apply the correct test. T student and mean+SD are only applicable for data that is normally distributed. Please check it.

Acceptable
